# A Comprehensive Study on Large-Scale Graph Training: Benchmarking and Rethinking

**Keyu Duan[1], Zirui Liu[2], Peihao Wang[3], Wenqing Zheng[3],**
**Kaixiong Zhou[2], Tianlong Chen[3], Xia Hu[2], Zhangyang Wang[3]**
[1]National University of Singapore, [2]Rice University, [3]University of Texas at Austin
{k.duan}@u.nus.edu; {zl105,Kaixiong.Zhou,xia.hu}@rice.edu;
{peihaowang,w.zheng,tianlong.chen,atlaswang}@utexas.edu

## Abstract

Large-scale graph training is a notoriously challenging problem for graph neural networks (GNNs). Due to the nature of evolving graph structures into the training process, vanilla GNNs usually fail to scale up, limited by the GPU memory space. Up to now, though numerous scalable GNN architectures have been proposed, we still lack a comprehensive survey and fair benchmark of this reservoir to find the rationale for designing scalable GNNs. To this end, we first systematically formulate the representative methods of large-scale graph training into several branches and further establish a fair and consistent benchmark for them by a greedy hyperparameter searching. In addition, regarding *efficiency*, we theoretically evaluate the time and space complexity of various branches and empirically compare them w.r.t GPU memory usage, throughput, and convergence. Furthermore, We analyze the pros and cons for various branches of scalable GNNs and then present a new ensembling training manner, named *EnGCN*, to address the existing issues. Remarkably, our proposed method has achieved new state-of-the-art (SOTA) performance on large-scale datasets. Our code is available at https://github.com/VITA-Group/Large_Scale_GCN_Benchmarking.

## 1 Introduction

The Graph Neural Networks (GNNs) have shown great prosperity in recent years [1–4], and have dominated a variety of applications, including recommender systems [5–7], social network analysis [8–10], scientific topological structure prediction (e.g. cellular function prediction [11, 12], molecular structure prediction [13, 14], and chemical compound retrieval [15]), and scalable point cloud segmentation [16, 17], etc. Although the message passing (MP) strategy provides GNNs' superior performance, the nature of evolving massive topological structures prevents MP-based GNNs [18–20, 1, 2, 4, 21, 22] from scaling to industrial-grade graph applications. Specifically, as MP requires nodes aggregating information from their neighbors, the integral graph structures are inevitably preserved during forward and backward propagation, thus occupying considerable running memory and time. For example [6], training a GNN-based recommendation system over 7.5 billion items requires three days on a 16-GPU cluster (384 GB memory in total).

To facilitate understanding, a unified formulation of MP with $K$ layers is presented as follows:

$$\mathbf{X}^{(K)} = \mathbf{A}^{(K-1)}\sigma\left(\mathbf{A}^{(K-2)}\sigma\left(\cdots\sigma(\mathbf{A}^{(0)}\mathbf{X}^{(0)}\mathbf{W}^{(0)})\cdots\right)\mathbf{W}^{(K-2)}\right)\mathbf{W}^{(K-1)}, \tag{1}$$

where $\sigma$ is an activation function (e.g. ReLU) and $\mathbf{A}^{(l)}$ is the weighted adjacency matrix at the $l$-th layer. As in Equ. (1), the key bottleneck of vanilla MP lies on $\mathbf{A}^{(l)}\mathbf{X}^{(l)}$. For the memory usage, the entire sparse adjacency matrix is supposed to be stored in one GPU. As the number of nodes grows, it is quite challenging for a single GPU to afford the message passing over the full graph.

Up to now, massive efforts have been made to mitigate the aforementioned issue and scale up GNNs [23, 24, 3, 25–30]. Most of them focus on approximating the iterative full-batch MP to reduce the memory consumption for training within one single GPU. It is worth noting that we target at the algorithmic scope and do not extend to scalable infrastructure topics like distributed training with multiple GPUs [31, 32] and quantization [33]. Briefly, the previous works encompass two branches: *Sampling-based* and *Decoupling-based*. Namely, the former methods [3, 25, 24, 23, 34–36] perform *batch training* that utilizes sampled subgraphs as a small batch to approximate the full-batch MP so that the memory consumption is considerably reduced. The latter follows the principle of performing *propagation* ($\mathbf{A}^{(l)}\mathbf{X}^{(l)}$) and *prediction* ($\mathbf{X}^{(l)}\mathbf{W}^{(l)}$) separately, either precomputing the propagation [27, 28, 19, 37, 31] or post-processing with label propagation [29, 38]. Despite the prosperity of scalable GNNs, there are still plights under-explored: we lack a systematic study of the reservoir from the perspective of *effectiveness* and *efficiency*, without which it is unachievable to tell the rationale of the designing philosophy for large-scale graph learning in practice.

To this end, we first establish a consistent benchmark and provide a systematic study for large-scale graph training for both *Sampling-based* methods (Sec. 2.1) and *Decoupling-based* methods (Sec. 2.2). For each branch, we conduct a thorough investigation of the design strategy and implementation details of typical methods. Then, we carefully examine the sensitive hyperparameters and unify them in one "sweet spot" set by a linear greedy hyperparameter (HP) search (Sec. 3), i.e., iteratively searching the optimal value for an HP while fixing the others. For all concerned methods, the performance comparison is conducted on representative datasets of different scales, varying from about $80,000$ nodes to $2,400,000$, including Flickr [24], Reddit [3], and ogbn-products [11]. This step is a crucial precondition on our way to the ultimate as the configuration inconsistency significantly prohibits a fair comparison as well as the following analysis. Besides, regarding *efficiency*, we theoretically present the time and space complexities for the various branches, and empirically evaluate them on GPU memory usage, throughput, and convergence (Sec. 4). In addition to the benchmark, we further present a new ensembling training manner *EnGCN* (Sec. 5) to address the existing issues mentioned in our benchmark analysis (Sec. 5.1). Notably, via organically integrating with self-label-enhancement (SLE) [29], EnGCN achieves the new state-of-the-art (SOTA) on multiple large-scale datasets.

## 2 Formulations For Large-scale Graph Training Paradigms

### 2.1 Sampling-based Methods

Given the formulation of Equ. (1), *sampling-based* paradigm seeks the optimal way to perform batch training, such that each batch will meet the memory constraint of a single GPU for message passing. For completeness, we restate the unified formulation of sampling-based methods as follows:

$$\mathbf{X}_{\mathcal{B}_0}^{(k)} = \widetilde{\mathbf{A}}_{\mathcal{B}_1}^{(k-1)} \sigma\left( \widetilde{\mathbf{A}}_{\mathcal{B}_2}^{(k-2)} \sigma\left( \cdots \sigma(\widetilde{\mathbf{A}}_{\mathcal{B}_K}^{(0)} \mathbf{X}_{\mathcal{B}_K}^{(0)} \mathbf{W}^{(0)}) \cdots \right) \mathbf{W}^{(K-2)} \right) \mathbf{W}^{(K-1)}, \tag{2}$$

where $\mathcal{B}_l$ is the set of sampled nodes for the $l$-th layer, and $\widetilde{\mathbf{A}}^{(l)}$ is the adjacency matrix for the $l$-th layer sampled from the full graph. Given the local view of GNN — one node's representation is only related to its neighbors — a straightforward way for unbiased batch training would be $\mathcal{B}_{l+1} = \mathcal{N}(\mathcal{B}_l)$, where $\mathcal{N}$ denotes the set of neighbors. $\mathcal{B}_0$ is randomly sampled according to the uniform distribution. Notably, this batch training style could achieve SOTA performance but also suffers from the "*neighbor explosion*" problem, where the time consumption and memory usage grow exponentially with the GNN depth, causing significant memory and time overhead. To mitigate this, a number of *sampling-based* methods were proposed. The key difference among them is how $\{\mathcal{B}_0, \ldots, \mathcal{B}_{K-1}, \mathcal{B}_K\}$ are sampled. Given a large-scale graph $\mathcal{G} = (\mathcal{V}, \mathcal{E})$, there are three categories of widely-used sampling strategies:

**Node-wise Sampling [3]**   $\mathcal{B}_{l+1} = \bigcup_{v \in \mathcal{B}_l} \{u \mid u \sim Q \cdot \mathbb{P}_{\mathcal{N}(v)}\}$, where $\mathbb{P}$ is a sampling distribution; $\mathcal{N}(v)$ is the sampling space, i.e., the 1-hop neighbors of $v$; and $Q$ denotes the number of samples. The representative node-wise sampling method is:

$\star$ *GraphSAGE* [3]: In GraphSAGE, $\mathbb{P}$ is the uniform distribution.

Compared with the aforementioned *naive batch training*, the node-wise sampling [3] alleviates the "*node explosion*" problem by fixing the number of sampled neighbors $Q$ for each node. It thus reduces the space complexity from $D^K$ to $Q^K$, where $D$ is the averaged node degree. However, as $Q$ is not far less than $D$ in order of magnitude, such mitigation is moderate, which is empirically validated by our empirical results in Sec. 3 and Sec. 4.

**Layer-wise Sampling [25, 26].** $\mathcal{B}_{l+1} = \{u \mid u \sim Q \cdot \mathbb{P}_{\mathcal{N}(\mathcal{B}_l)}\}$, where $\mathcal{N}(\mathcal{B}_l) = \bigcup_{v \in \mathcal{B}_l} \mathcal{N}(v)$ denotes the union of 1-hop neighbors of all nodes in $\mathcal{B}_l$. We introduce a couple of layer-wise sampling methods as follows.

⋆ *FastGCN* [25]: The sampling distribution $\mathbb{P}$ is designed regarding the node degree, where the probability for node $u$ of being sampled is $p(u) \propto ||\hat{\mathbf{A}}(u, :)||^2$.

⋆ *LADIES* [26]: More recently, based on FastGCN, Zou et.al. [26] propose LADIES that extends the sampling space from $\mathcal{N}(\mathcal{B}_l)$ to $\mathcal{N}(\mathcal{B}_l) \cup \mathcal{B}_l$ by adding the self-loops.

Notably, Compared with the node-wise sampling, the layer-wise sampling essentially solves the "*neighbor explosion*" problem by fixing the number of overall sampled nodes in a layer to $Q$. However, the layer-wisely induced adjacency matrix is usually sparser than the others, which accounts for its sub-optimal performance in practice.

**Subgraph-wise Sampling [23, 24].** $\mathcal{B}_K = \mathcal{B}_{K-1} = \cdots = \mathcal{B}_0 = \{u \mid u \sim Q \cdot \mathbb{P}_{\mathcal{G}}\}$. In the subgraph-wise sampling, all layers share the same subgraph induced from the entire graph $\mathcal{G}$ based on a specific sampling strategy $\mathbb{P}_{\mathcal{G}}$, such that the sampled nodes are confined in the subgraph. Typically, this sampling strategy has two representative works:

⋆ *ClusterGCN* [23]: ClusterGCN first partitions the entire graph into clusters based on some graph partition algorithms, e.g. METIS [39], and then select several clusters to form a batch.

⋆ *GraphSAINT* [24]: GraphSAINT samples a subset of nodes based on sampling strategy $\mathbb{P}_{\mathcal{G}}$ and then induces the corresponding subgraph as a batch. The commonly-used sampling strategies include: $(i)$ node sampler: $\mathbb{P}(u) = ||\widetilde{\mathbf{A}}_{:,u}||^2$, $(ii)$ edge sampler: $\mathbb{P}(u, v) = \frac{1}{deg(u)} + \frac{1}{deg(v)}$, and $(iii)$ random walk sampler. They are illustrated in Appendix A1.1.

## 2.2 Decoupling-based Methods

Training GNNs with full-batch message passing at each epoch is not plausible. In this section, we summarize another line of scalable GNNs which decouple the message passing from GPU training to CPUs. Specifically, the message passing is conducted only once at CPUs accompanied by large accessible memory. Depending on the processing order, there are two typical ways to decouple these two operations: $(i)$ *pre-processing* and $(ii)$ *post-processing*.

**Pre-processing: MP precomputating [27–29].** Recalling Equ. (1), without loss of generalization, we assume that $\mathbf{A}^{(k-1)} = \mathbf{A}^{(k-2)} = \cdots \mathbf{A}^{(0)} = \mathbf{A}$, i.e. the topological structure for the entire graph remains the same during forward propagation, meeting most of the cases. To decouple the two operations, *message passing* ($\mathbf{AX}$) and *feature transformation* ($\mathbf{XW}$), we can first pre-compute the propagated node representations and then train a neural network for the downstream task based on these fused representations:

$$\underbrace{\mathbf{X}^l = \mathbf{A}^l \mathbf{X}}_{precomputing}, \quad \underbrace{\bar{\mathbf{X}} = \rho(\mathbf{X}, \mathbf{X}^1, \cdots, \mathbf{X}^K), \quad \mathbf{Y} = f_\theta(\bar{\mathbf{X}})}_{end\text{-}to\text{-}end\ training\ on\ a\ GPU}, \tag{3}$$

where $\mathbf{X}^l$ can be regarded as the node representation aggregating $l$-hop neighborhood information; $K$ is the largest propagation hop; $\rho(\cdot)$ is a function that combines the aggregated features from different hops; and $f_\theta(\cdot)$ is a feature mapping function parameterized by $\boldsymbol{\theta}$. We summarize three existing pre-computing schemes as follows.

⋆ *SGC* [27]: SGC leverages the node representations aggregated with k hops and feeds the resultant features to a full-connected layer. We can formulate this scheme by letting $\rho(\cdot)$ select the last element $\mathbf{X}^K$ and $f_\theta(\cdot)$ be a linear layer with readout activation: $\mathbf{Y} = \sigma(\mathbf{X}^K \mathbf{\Theta})$.

⋆ *SIGN* [28]: SIGN concatenates features from different hops and then fuse them as the final node representation via a linear layer. To be more specific, $\rho(\cdot)$ is defined as $\bar{\mathbf{X}} = \begin{bmatrix} \mathbf{X} & \mathbf{X}^1 & \cdots & \mathbf{X}^K \end{bmatrix} \mathbf{\Omega}$, where $\mathbf{\Omega}$ is a transformation matrix, and $f_\theta(\cdot)$ is defined as a linear readout layer $\mathbf{Y} = \sigma(\bar{\mathbf{X}} \mathbf{\Theta})$.

⋆ *SAGN* [29]: SAGN adopts attention mechanism to combine feature representations from $K$ hops: $\bar{\mathbf{X}} = \sum_{l=1}^K \mathbf{T}^l \mathbf{X}^l$, where $\mathbf{T}^l$ is a diagonal matrix whose diagonal corresponds to the attention weight for each node of $k$-hop information. The attention weight for the $i$-th node is calculated by $T_i^k = \text{softmax}_K(\text{LeakyReLU}(\boldsymbol{u}^T \mathbf{X}_i + \boldsymbol{v}^T \mathbf{X}_j^k))$, where the subscripts slices the data matrices along the row. The feature mapping function is implemented by an MLP block with a skip connection to initial features: $\mathbf{Y} = \text{MLP}_\theta(\bar{\mathbf{X}} + \mathbf{X}\mathbf{\Theta}_r)$.

**Post-processing: Label Propagation.** The label propagation algorithms [40–44, 38, 45] diffuse labels in the graph and make predictions based on the diffused labels. It is a classical family of graph algorithms for *transductive learning*, where the nodes for testing are used in the training procedure. The label propagation can be written in a unified form as follows:

$$\mathbf{Y}^{(l)} = \alpha \mathbf{A} \mathbf{Y}^{(l-1)} + (1 - \alpha) \mathbf{G}. \tag{4}$$

The diffusion procedure iterates the formula above with $l$ for multiple times to guarantee convergence. It requires two sets of inputs: $(i)$ the stack of the *label embeddings* of all nodes, denoted as $\mathbf{Y}^{(0)} \in \mathbb{R}^{N \times c}$, where $c$ is the number of classes. In our implementation, the $\mathbf{Y}^{(0)}$ is the output of a trained MLP model [38]. $(ii)$ the *diffusion embedding*, denoted as $\mathbf{G} \in \mathbb{R}^{N \times c}$ that propagate themselves across the edges in the graph. Depending on how the diffusion embeddings of unlabeled nodes are computed, two types of $\mathbf{G}$ are summarized as follows:

⋆ *Zeros* [40]: $\mathbf{G}_{i,:} = \begin{cases} \hat{\mathbf{Y}}_{i,:} - \alpha \mathbf{A} \mathbf{Y}^{(k)}_{i,:}, & i \in \mathcal{T}_{train} \\ \mathbf{0}, & otherwise \end{cases}$, where $\mathcal{T}_{train}$ denotes the training set and $\hat{\mathbf{Y}}$ is the stack of true labels. For *zeros*, $\mathbf{Y}^{(0)} = \mathbf{G}$.

⋆ *Residual* [38]: $\mathbf{G}_{i,:} = \begin{cases} \hat{\mathbf{Y}}_i, & v_i \in \mathcal{T}_{train} \\ \hat{\mathbf{Z}}_i, & otherwise \end{cases}$, where $\hat{\mathbf{Z}} = \mathbf{Z} + \hat{\mathbf{E}}$. $\mathbf{Z}$ is the predictions of a trained simple neural network, e.g. MLP, and $\hat{\mathbf{E}}$ is an residual error matrix, which is optimized iteratively for multiple times by $\mathbf{E}^{t+1} = (1 - \alpha)\mathbf{E} + \alpha \mathbf{A} \mathbf{E}^{(t)}$, where $\mathbf{E} = \mathbf{Z} - \hat{\mathbf{Y}}$ and $\mathbf{E}^{(0)} = \mathbf{E}$.

### 2.3 More Related Works

**Model-agnostic Tricks.** Besides the training methods as introduced above, there are some model-agnostic tricks that have been empirically confirmed to be effective for boosting large-scale graph training. Although those add-ons cannot be included into our benchmarking analysis, it is of equal importance to introduce them for completeness. Here we briefly introduce two representative ones:

⋆ *Self-Label-Enhanced (SLE)* [29]: SLE includes two individual tricks, *self training* and *label augmentation*. Here we use $\mathcal{T}$ denoting the training set. For self training, the unlabeled nodes with high confidence (larger than a pre-defined threshold) are added to $\mathcal{T}$ after a certain number of training epochs. For label augmentation, it trains an additional model $\Phi(\cdot)$. The forward propagation can be formulated as $out = \Phi(\hat{\mathbf{A}}^k \mathbf{Y}_{\mathcal{T}})$. $out$ is added to the main model to make the final prediction.

⋆ *GIANT* [46]: In general, node features are usually pre-embed with graph-agnostic language models, such as word2vec [47] and BERT [48]. Recently, Chien et.al. propose a graph-related node feature extraction framework (GIANT), which embeds the raw texts to numerical features by taking advantage of graph structures, to help boost the performance of GNNs for the downstream tasks.

**Memory-based GNN Training**. Focus on mitigating the "*Neighbor Explosion*" problem of full-batch training as introduced, memory-based GNNs [49, 50] try to save the GPU memory with different techniques while including all neighbor nodes into computing during the message passing. GAS [49] incorporates *historical embeddings* [34] to provably maintain the expressive power of full-batch GNN. VQ-GNN [50] utilizes *vector quantization* to scale convolutional-based GNN and resemble the performance of full-batch message passing by learning an additional quantized feature matrix and a corresponding low-rank adjacent matrix.

## 3 Benchmarking Over Effectiveness

### 3.1 Implementation Details

We test numerous large-scale graph training methods with a greedy hyperparameter (HP) search to find their *sweet spot* and the best performance for a fair comparison. The search space is defined in Table 1. The access and statistics of all used datasets are introduced in Appendix A3.1. Particularly, for label propagation, we select two representative algorithms: Huang et.al. [38], the *residual* diffusion type, and Zhu et.al. [40], the *zeros* type. The number of propagation is the maximum iteration $k$. The aggregation ratio is $\alpha$ as in Equ. (4), and the number of MLP layers is the number of MLP layers that precedes the label propagation module following Huang et.al. [38].

Limited by space, we select five representative approaches that covers all branches as we introduced, including GraphSAGE [3], LADIES [26], ClusterGCN [23], SAGN [29], and C&S [38]. We illustrate the selected results in Fig. 1 and the results of other methods in Fig. A5. For each subplot, from left to right, each column denotes the search results for one HP. Once one HP was searched, its value will be fixed to the best results for the rest HP searching. Iteratively, we obtain the best performance in the last column. For convenience and clarity, we list the searched optimal hyperparameter settings of all test methods in Table A5.

Table 1: The search space of hyperparameters for benchmarked methods.

| Category | Hyperparameter (Abbr.) | Candidates |
|---|---|---|
| Sampling & Precomputing | Learning rate (LR) | $\{1e-2^*, 1e-3, 1e-4\}$ |
| | Weight Decay (WD) | $\{1e-4^*, 2e-4, 4e-4\}$ |
| | Dropout Rate (DP) | $\{0.1, 0.2^*, 0.5, 0.7\}$ |
| | Training Epochs (#E) | $\{20, 30, 40, 50^*\}$ |
| | Hidden Dimension (HD) | $128^*, 256, 512$ |
| | # layers (#L) | $\{2^*, 4, 6\}$ |
| | Batch size$^a$ (BS) | $\{1000^*, 2000, 5000\}$ |
| LP | Diffusion Type (DT) | { residual$^*$, zeros } |
| | # Propagations (#Prop) | { 2, 20$^*$, 50 } |
| | Aggregation Ratio (AR) | { 0.5, 0.75$^*$, 0.9, 0.99 } |
| | Adj. Norm (Adj.) | { $\mathbf{D}^{-1}\mathbf{A}$, $\mathbf{A}\mathbf{D}^{-1}$, $\mathbf{D}^{-1/2}\mathbf{A}\mathbf{D}^{-1/2*}$ } |
| | Auto Scale (AS) | { $True^*$, $False$ } |
| | # MLP Layers (#ML) | { 2$^*$, 3, 4 } |

$^*$ marks the default value
$^a$ we do not search batch size for precomputing based methods since they do not follow a sample-training style.

## 3.2 Experimental Observations

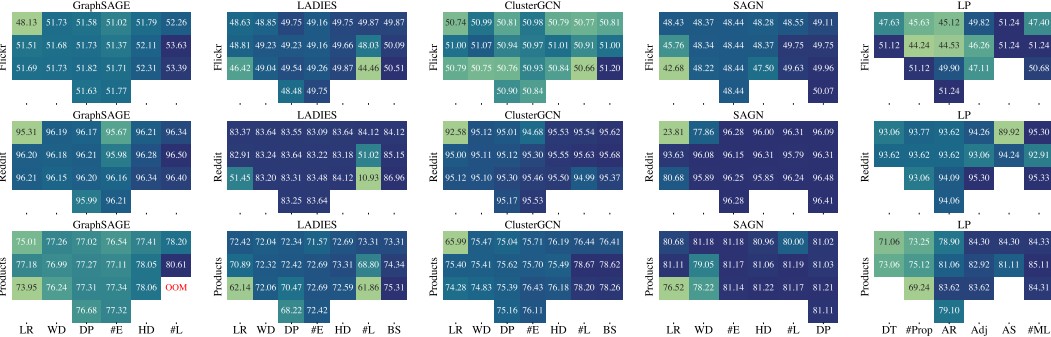

Figure 1: The greedy hyperparameter searching results for selected representative methods. The x-axis denotes the searched HPs, where the abbreviations are consistent with Table 1.

**Obs. 1. Sampling-based methods are more sensitive to the hyperparameters related to MP.** According to Fig. 1, in comparison with precomputing, all sampling-based methods are non-sensitive to hyperparameters (HPs) that are related to the feature transformation matrices, including weight decay, dropout, and hidden dimension; but particularly sensitive to the MP-related HPs, including the number of layers and batch size. For model depth, sampling-based methods generally achieve the *sweet spots* when the number of layers is confined to shallow and suffers from the *oversmoothing* problem [51–55] as the GNN models go deeper. However, this issue is moderately mitigated in decoupling-based methods as the model depth does not align with the number of MP hop.

**Obs. 2. Sampling-based methods' performance is nearly positive-correlated with the training batch size.** According to the results of the last column of all sampling-based methods, the performance of the layer-wise and subgraph-wise sampling methods is roughly proportional to the batch size. Expectedly, the model performance could further increase as the batch size grows till the upper bound of full-batch training because more links can be preserved. Particularly, in our experiment, we set the number of sampled neighbors of *node-wise sampling* to a large threshold such that the performance of GraphSAGE can be regarded as *full-batch training*'s. It can be easily found that the performance of sampling-based methods is inferior to *full-batching training* (GraphSAGE), further proving our conjecture that the missing links by sampling are non-trivial.

**Obs. 3. Precomputing-based methods generally perform better on larger datasets.** As show in Fig. 1 and Fig. A5, C&S (*label propagation*) outperforms the *full-batch training* (GraphSAGE as introduced in *Obs. 2*) on the largest dataset ogb-products by a large margin of 4.5%, although both two branches have on-par performance on smaller datasets. Remarkably, our searched results for GraphSAGE and LP on ogbn-products also reached better performance, compared with the ones on the OGB leaderboard [1]. Noticing that GraphSAGE encounters the out-of-memory (OOM)[2] runtime

[1] https://ogb.stanford.edu/docs/leader_nodeprop/
[2] We rerun it on a GPU with larger memory and the accuracy is 80.56%

error with increasing depth, the observation partially indicates that, limited by model depth and *neighbor explosion* problem, it is possibly not powerful for extremely large-scale graphs to learn expressive representations.

## 4 Benchmarking Over Efficiency

### 4.1 Time And Space Complexity

In this section, we present another benchmark regarding the efficiency of scalable graph training methods. Firstly, we briefly summarize a general complexity analysis in Table 2. For *sampling-based* methods, we note that the time complexity is for training GNNs by iterating over the whole graph. The time complexity $\mathcal{O}(L||\mathbf{A}||_0 D + LND^2)$ consists of two parts. The first part $L||\mathbf{A}||_0 D$ is from the Sparse-Dense Matrix Multiplication,i.e., $\mathbf{AX}$. The second part $LND^2$ is from the normal Dense-Dense Matrix Multiplication, i.e., $(\mathbf{AX})\mathbf{W}$. Regarding the space complexity, we need to store the activations of each layer in memory, which has a $\mathcal{O}(bLD)$ space complexity. Note that we ignore the memory usage of model weights and the optimizer here since they are negligible compared to the activations. For *decoupling-based* methods, the training paradigm is simplified as MLPs, and thus the complexity is the same as the traditional mini-batch training. We do not include *label propagation* in our analysis since it can be trained totally on CPUs.

### 4.2 Throughput And Memory Usage

**Implementation Details**    To fairly benchmark the training speed and memory usage for large-scale graph training methods, we empirically evaluate the throughputs and actual memory for various methods during the training procedure. Here "Throughput" measures how many times can we complete the training steps within a second. Note that we omit the label propagation methods since it is not

Table 2: The time and space complexity for training GNNs with sampling-based and decoupling-based methods, where $b$ is the averaged number of nodes in the sampled subgraph and $r$ is the averaged number of neighbors of each node. Here we do not consider the complexity of pre-processing sice it can be done in CPUs.

| Category | Time Complexity | Space Complexity |
|---|---|---|
| Node-wise Sampling [3] | $\mathcal{O}(r^L ND^2)$ | $\mathcal{O}(br^L D)$ |
| Layer-wise Sampling [28, 26] | $\mathcal{O}(rLND^2)$ | $\mathcal{O}(brLD)$ |
| Subgraph-wise Sampling [23, 24] | $\mathcal{O}(L||\mathbf{A}||_0 D + LND^2)$ | $\mathcal{O}(bLD)$ |
| Precomputing [27–29] | $\mathcal{O}(LND^2)$ | $\mathcal{O}(bLD)$ |

trained by *backward propagation*. We provide our implementation details for computing the throughput and memory usage in section A3.2. We report the hardware throughput and activation usage in Table 3. We summarize three main observations.

Table 3: The memory usage of activations and the hardware throughput (higher is better). The hardware here is an RTX 3090 GPU.

| | Flickr | | Reddit | | ogbn-products | |
|---|---|---|---|---|---|---|
| | Act | Throughput | Act | Throughput | Act | Throughput |
| | Mem. (MB) | (iteration/s) | Mem. (MB) | (iteration/s) | Mem. (MB) | (iteration/s) |
| GraphSAGE | 230.63 | 65.96 | 687.21 | 27.62 | 415.94 | 37.69 |
| ClusterGCN | 18.45 | 171.46 | 20.84 | 79.91 | 10.62 | 156.01 |
| GraphSAINT | 16.51 | 151.77 | 21.25 | 70.68 | 10.95 | 143.51 |
| FastGCN | 19.77 | 226.93 | 22.53 | 87.94 | 11.54 | 93.05 |
| LADIES | 33.26 | 195.34 | 43.21 | 116.46 | 20.33 | 93.47 |
| SGC | 0.01 | 115.02 | 0.02 | 89.91 | 0.01 | 267.31 |
| SIGN | 16.99 | 96.20 | 16.38 | 75.33 | 16.21 | 208.52 |
| SAGN | 72.94 | 55.28 | 72.37 | 43.45 | 71.81 | 80.04 |

**Obs. 4. GraphSAGE is significantly slower and occupies more memory compared to other baselines.** This is partially because of the large neighbor sampling threshold we set and inherently owing to its neighborhood explosion. Namely, to compute the loss for a single node, it requires the neighbors' embeddings at the down-streaming layer recursively. Please refer to Sec. 2.1 for details.

**Obs. 5. SGC does not occupy any activation memory.** As shown in Table 3, SGC only occupies about 0.01 MB actual memory during training. This is because SGC only has one linear layer and the activation is exactly the input feature matrix, which has been stored in memory. Thus, it is not accounted towards the activation memory.

**Obs. 6. In general, the speed of decoupling-based methods is comparable to sampling-based methods.** Besides the scale of sparse adjacency matrix, the feature set size is also crucial for

occupying memory. Although precomputing-based methods avoid storing the graph structures in a GPU, they may take advantage of multi-hop features, where the corresponding memory is multiplied many times.

## 4.3 Convergence Analysis

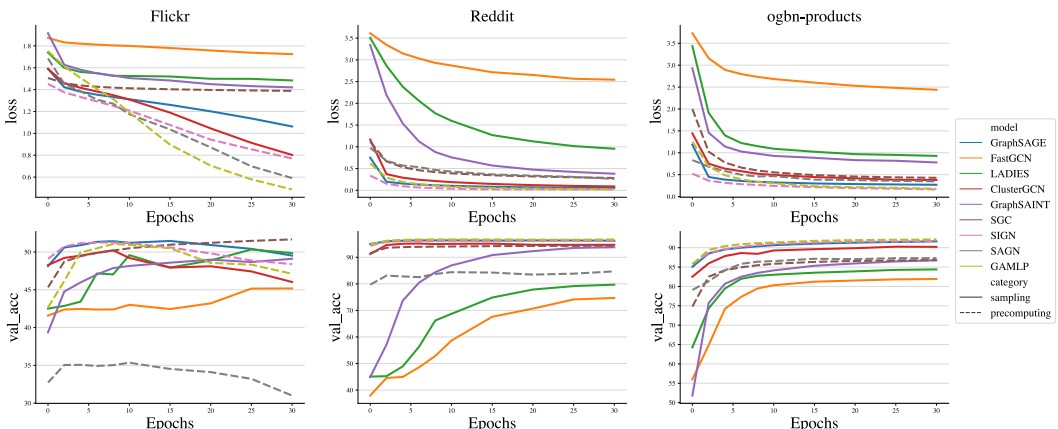

Figure 2: The empirical results of convergence for *sampling-based* methods (real line) and *precomputing-based* methods (dash line).

For convergence analysis, we test all benchmarked methods on Flickr, Reddit, and ogbn-products. The training loss and validation accuracy (val_acc) are shown in Fig. 2. Based on the empirical results, we summarize the main observation as follows:

**Obs. 7. In general, precomputing-based methods have faster and more stable convergence than sampling-based methods.** This is because sampling-based methods usually incur a variance among batches that poses unstable and slow convergence [35]. However, precomputing-based methods mitigate this by moving message passing from backward propagation to the precomputing stage.

# 5 EnGCN: Rethinking Graph Convolutional Networks With Ensembling

## 5.1 An Empirical Summary: Pros And Cons

Based on our benchmark results in section 3 and 4, we summarize the advantages (marked as **Pros**) and constraints (marked as **Cons**) for different branches as follows. Besides the summary, we also provide a joint comparison of effectiveness and efficiency for various methods in Appendix A2.2.

⋆ *Sampling-based*: (**Pros**) Sampling subgraphs into GPU training allows them taking advantage of numerous graph convolution layers, such as GCN [1], GraphSAGE [3], and GIN [4], and this is flexible for them to design specific architecture for different downstream tasks, e.g. node classification and graph classification. (**Cons**) As aforementioned, sampling-based methods suffers from *link sparsity* [24] (section 3) and unstable and low convergence (section 4) problems, both of which prevent sampling-based methods from achieving SOTA performance.

⋆ *Precomputing-based*: (**Pros**) Decoupling *message passing* from GPU training to CPU precomputing allows precomputing-based methods utilize a mixture of feature transformation units (e.g. attention mechanism and MLP) to train in a well-studied manner. This guarantees stable and fast convergence (section 4). Particularly, integrating with the add-ons like SLE [29] and GIANT [46], precomputing-based methods achieve SOTA performance on large-scale *open graph benchmark* (ogb) [11] datasets. (**Cons**) In general, precomputing-based methods at least occupy a CPU memory space of $\mathcal{O}(LNd)$, where $L$ is the number of layers; $N$ is the number of nodes; and $d$ is the dimension of input features. In comparison, it is $L$ times as large as the others, which is not affordable for extremely large-scale graphs. For example, containing about 111 million nodes, the largest ogb dataset, ogbn-papers100M, requires approximately 57 Gigabytes (GB) to store the initial feature matrix, given the data type is float and the dimension of features is 128. As the number of layers increases, the required CPU memory space will grow proportionally to an unaffordable number.

⋆ *Label Propagation*: (***Pros***) As a traditional branch of graph learning algorithm, label propagation is a simple but effective add-on as a post-processing trick nowadays. Because of its mode-agnostic nature, it can be simply attached to the end of any graph representation learning algorithm to boost the final prediction. (***Cons***) Label propagation has many additional sensitive hyperparameters as we introduced in Table 1 and is specifically designed for the node classification task.

## 5.2 Motivation and Related Works

To address the above constraints of *sampling-based* methods and *precomputing-based* methods, let us first recap the full-batch message passing in Equ. (1). We reformulate it into a more general form:

$$\mathbf{X}^{(k)} = \Phi^{(k-1)}\bigg(\mathbf{A}\Phi^{(k-2)}\big(\cdots \mathbf{A}\Phi^{(0)}(\mathbf{A}\mathbf{X}^{(0)})\big)\bigg),$$

where $\Phi^{(i)}$ denotes the feature mapping model for the $i$-th layer. To make the message passing scalable, following the rationale of $decoupling$, we propose a different training scheme from precomputing: *Instead of end-to-end training, we sequentially train the $\Phi$s in a layer-wise manner.* In this way, no precomputing is required and thus the corresponding constraint of CPU memory occupation is essentially mitigated. To elaborate on this, we present the layer-wise training manner:

$$\underbrace{\mathbf{X}^{(l)} = \mathbf{A}\mathbf{X}^{(l-1)}}_{\textit{Message passing on CPUs}}, \quad \underbrace{\mathbf{Z}^{(l)} = \Phi^{(l)}(\mathbf{X}^{(l)})}_{\textit{forword propagation}}, \quad \underbrace{\nabla\Phi^{(l)} = \nabla\mathcal{L}(\mathbf{Z}^{(l)}, \mathbf{Y})}_{\textit{backward propagtion}}. \tag{5}$$

From layer 0 to $k$, we do message passing once and then train $\Phi^{(l)}$ in batches for epochs. Finally, one can simply use the output of model $\Phi$ as the prediction. Besides, from the perspective of ensembling, the models $\Phi$ can be naturally viewed as a set of weak learners trained on multiple views of the input $\mathbf{X}$. As a result, it is compatible to use ensembling to boost the final predictions, such as majority voting. In addition, Based on our empirical results, this training manner is capable of achieving SOTA methods on relatively small datasets without exhaustive finetuning. To further boost the performance, we organically integrate SLE, which *has achieved new SOTA performance on several representative datasets*. We name this model *EnGCN* (Ensembling GCN).

**Related Works.** Interestingly, the layer-wise training manner and majority voting are naturally consistent with the *boosting* algorithms, where we sequentially train weak learners with instance reweighting and make the final prediction by majority voting. In the scope of graph representation learning, AdaGCN [56] first applies adaboosting [57, 58] to address the *oversmoothing* problem of deep GCNs. Though focusing on different topics, AdaGCN has a similar training scheme as ours. Therefore, we implement a scalable version for it, which is included as a SOTA baseline in our experiment. In addition, AdaClusterGCN [59] proposed an adaboosting application that ensembles weak learners trained on different clusters.

## 5.3 Methodology

Considering a large-scale graph $\mathcal{G} = (\mathbf{A}, \mathbf{X}, \mathbf{y})$, where $\mathbf{A}$ is the adjacent matrix, $\mathbf{X}$ is the node features, and $\mathbf{y}$ is the true labels. Respectively, $\mathcal{T}_{train}$, $\mathcal{T}_{val}$ and $\mathcal{T}_{test}$ denotes the training, validation, and test set. Let $\mathbf{X}^{(l)}$ and $\mathbf{Y}^{(l)}$ denotes the embeddings of node features and labels at the $l$-th layer, respectively. We use $\tilde{\mathbf{y}}^{(l)}$ and $\widetilde{\mathcal{T}}^{(l)}_{train}$ denoting the pseudo labels, pseudo training set for self training at layer $l$.

**Initialization.** we initialise several important matrices and vectors:

$$\mathbf{X}^{(0)} = \mathbf{X}, \quad \mathbf{Y}^{(0)}_{i,:} = \begin{cases} \text{one\_hot}(\mathbf{y}_i), & i \in \mathcal{T}_{train} \\ \mathbf{0}, & \text{otherwise} \end{cases}, \quad \widetilde{\mathcal{T}}^{(0)}_{train} = \mathcal{T}_{train}, \quad \tilde{\mathbf{y}}^{(0)}_i = \begin{cases} \mathbf{y}_i, & i \in \widetilde{\mathcal{T}}_{train} \\ \mathbf{0}, & \text{otherwise} \end{cases}$$

**Layer-wise Training.** From 0 to $k$, we follow a layer-wise training manner, where each training stage contains three phases: *pre-processing*, *training*, and *post-processing*. For layer $l$, the three phases are introduced as follows.

*Pre-processing.* For pre-processing, we precompute $\mathbf{X}^{(l)}$ and $\mathbf{Y}^{(l)}$ in CPUs as follows:

$$\mathbf{X}^{(l)} = \hat{\mathbf{A}}\mathbf{X}^{(l-1)}, \quad \mathbf{Y}^{(l)} = \hat{\mathbf{A}}\mathbf{Y}^{(l-1)}, \tag{6}$$

where $\tilde{\mathbf{A}}$ is symmetrically normalized [1]. Note that pre-processing is skipped when $l = 0$.

*Training.* We solely train two simple models till convergence, which empirically takes dozens of epochs on real-world datasets. The forward propagation is:

$$\text{out}^{(l)} = \Omega(\mathbf{X}^{(l)}, \mathbf{Y}^{(l)}) = \Phi(\mathbf{X}^{(l)}) + \Psi(\mathbf{Y}^{(l)}), \tag{7}$$

where $\Phi$ and $\Psi$ are two MLP models that are shared through all layers. Specifically, when $l = 0$, the forward propagation is reduced to $\text{out}^{(0)} = \Phi(\mathbf{X}^{(0)})$ where $\Psi$ is not evolved. This is because the initialized $\mathbf{Y}^{(0)}$ contains many zero vectors and will pose the overfitting problem. For backward propagation, we compute the training loss using the pseudo labels $\tilde{\mathbf{y}}^{(l)}$ instead of $\mathbf{y}^{(l)}$.

*post-processing.* After obtaining the trained models, we save the state of them as $\Omega^{(l)} = (\Phi^{(l)}, \Psi^{(l)})$ for ensembling. Furthermore, *self training* is used to enhance the training set. Following Sun et.al. [29], the pseudo labels and pseudo training masks are updated as follows.

$$\widetilde{\mathcal{T}}_{train}^{(l+1)} = \widetilde{\mathcal{T}}_{train}^{(l)} \cup \{i \mid \max_c(\tau(\text{out}_i^{(l)})) \geq \alpha\}, \quad \tilde{\mathbf{y}}_i^{(l+1)} = \begin{cases} \mathbf{y}_i, & i \in \mathcal{T}_{train} \\ c, & \text{else if } \max_c(\tau(\text{out}_i^{(l)})) \geq \alpha \end{cases}, \tag{8}$$

where $\tau$ is the softmax function.

**Inference With Majority Voting.** After k layers, we have obtained a series of weak learners $\{\Omega^{(l)} \mid 0 \leq l \leq k\}$. The final prediction of node $n$ is made by weighted majority voting [58]:

$$\hat{y}_n = \underset{c}{argmax} \sum_{l=0}^{k} \left( \mathbf{z}_n^{(l)} - \frac{1}{d} \sum_{i=1}^{d} (\mathbf{z}_{n,i}^{(l)}) \right), \tag{9}$$

where $\mathbf{z}^{(l)} = log\_softmax(\text{out}_n^{(l)})$.

### 5.4 Empirical Analysis

**Experiment Settings.** Consistent with our effectiveness benchmark, we test our proposed *EnGCN* on Flickr, Reddit, and ogbn-products. A similar hyperparameter (HP) search was conducted to find its suitable HP setting. The search space is provided in Appendix A2.3. For the baselines, we directly use all benchmark results from section 3, where the SOTA performance has been achieved.

**Main Experiment.** As shown in Table 4, *EnGCN* outperforms the *sampling-based* and *decoupling-based* methods on multi-scale datasets. For Flickr and Reddit, EnGCN outperforms the baselines by a large margin. Remarkably, EnGCN has achieved new SOTA performance on ogbn-products, outperforming C&S by 2.88% and the SOTA model (GIANT-XRT+SAGN+MCR+C&S) in the ogb leaderboard by 1.26%. In addition to the comparison experiment, we also conduct a couple of ablation studies to provide more insights into EnGCN in Appendix A2.4.

Table 4: The comparison experiment results on Flickr, Reddit, and ogbn-products

| Category | Baselines | Flickr | Reddit | ogbn-products |
|---|---|---|---|---|
| Sampling-based | GraphSAGE [3] | $53.63 \pm 0.13\%$ | $96.50 \pm 0.03\%$ | $80.61 \pm 0.16\%$ |
| | FastGCN [25] | $50.51 \pm 0.13\%$ | $79.50 \pm 1.22\%$ | $73.46 \pm 0.20\%$ |
| | LADIES [26] | $50.51 \pm 0.13\%$ | $86.96 \pm 0.37\%$ | $75.31 \pm 0.56\%$ |
| | ClusterGCN [23] | $51.20 \pm 0.13\%$ | $95.68 \pm 0.03\%$ | $78.62 \pm 0.61\%$ |
| | GraphSAINT [24] | $51.81 \pm 0.17\%$ | $95.62 \pm 0.05\%$ | $75.36 \pm 0.34\%$ |
| Decoupling-based | SGC [27] | $50.35 \pm 0.05\%$ | $93.51 \pm 0.04\%$ | $67.48 \pm 0.11\%$ |
| | SIGN [28] | $51.60 \pm 0.11\%$ | $95.95 \pm 0.02\%$ | $76.85 \pm 0.56\%$ |
| | SAGN [29] | $50.07 \pm 0.11\%$ | $96.48 \pm 0.03\%$ | $81.21 \pm 0.07\%$ |
| | GAMLP [30] | $52.58 \pm 0.12\%$ | $96.73 \pm 0.03\%$ | $83,76 \pm 0.19\%$ |
| | C&S [38] | $51.24 \pm 0.17\%$ | $95.33 \pm 0.08\%$ | $85.11 \pm 0.07\%$ |
| Other SOTA Methods | AdaGCN [56] | $52.97 \pm 0.01\%$ | $96.05 \pm 0.00\%$ | $76.41 \pm 0.00\%$ |
| | SAGN+SLE [29]* | $54.60 \pm 0.40\%$ | $97.10 \pm 0.00\%$ | $84.28 \pm 0.14\%$ |
| | GIANT-XRT+ SAGN+MCR+C&S [60]* | - | - | $86.73 \pm 0.08\%$ |
| Ours | EnGCN | $\mathbf{56.43 \pm 0.21\%}$ | $\mathbf{97.14 \pm 0.03\%}$ | $\mathbf{87.99 \pm 0.04\%}$ |

∗: the results are from the original papers

**The Training Efficiency and Convergence Landscape of EnGCN.** For EnGCN, since all we need to train is two simple shallow MLPs (Section 5.3), the GPU throughput and memory consumption

are expected to be sufficiently efficient. The remained concern is solely about the convergence of EnGCN. Due to the nature of *layer-wise training*, the convergence of EnGCN is more complicated than other end-to-end training methods. In Figure 3, we show the convergence landscape of EnGCN and provide several interesting observations as follows.

❶ As shown in Figure 3, the train accuracy and validation accuracy generally increase layer-wisely till convergence. Noticeably, though the training accuracy occasionally drops, the validation accuracy still relatively remains positive. ❷ At the beginning of each layer, the accuracy changes rapidly, indicating the remarkable distribution difference for various hops. ❸ Different datasets are sensitive to different hops. For example, the 2-nd hop is crucial to boost the training and validation accuracy on Flickr, while for Reddit and ogbn-products, 1-hop neighbors are more important.

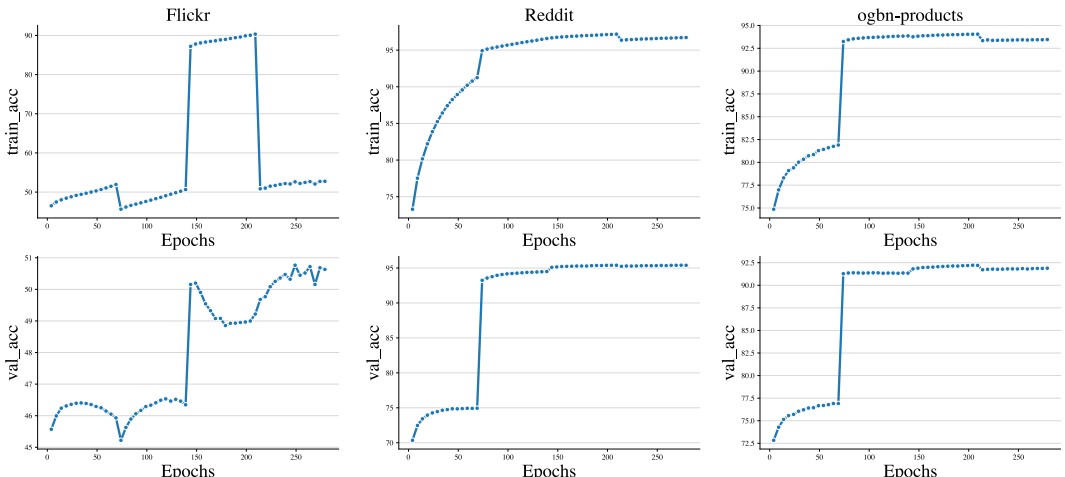

Figure 3: The convergence landscape of EnGCN. All models are trained with 4 layers' features. For each layer-wise phase, we train the model with 70 epochs.

**The CPU memory consumption of EnGCN.** To confirm the low CPU memory consumption of En-GCN, we provide a comparison experiment and illustrate the results in Figure 4. The x-axis denotes the models' number of layers while the y-axis records their allocated memories that are reported by "*aten::empty*" of PyTorch. As shown in Figure 4, the precomputing-based methods, SIGN and SAGN, suffer from expensive CPU memory consumption as the model depth increases. For sampling-based methods, since there is no need to pre-store a large number of feature matrices, the memory consumption increases much more smoothly. For EnGCN, as no precomputing is re-

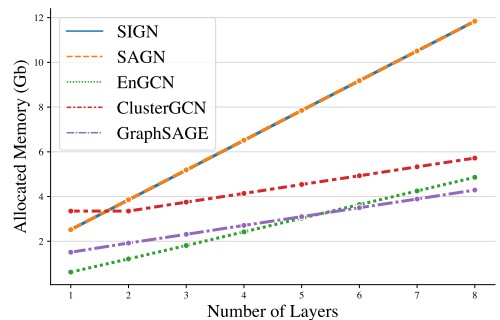

Figure 4: The allocated CPU memory of EnGCN and selected baselines on Flickr.

quired, the CPU memory consumption is considerably reduced in comparison with SIGN and SAGN, which intuitively validates the CPU memory efficiency of EnGCN.

## 6   Conclusion

The scalability issue of graph convolutional networks has been a notoriously challenging research problem. In this work, we establish a fair and consistent benchmark for large-scale graph training w.r.t effectiveness and efficiency. We provide a unified formulation for dozens of works and further assess them on the basis of accuracy, memory usage, throughput, and convergence. Furthermore, provided with the comprehensive benchmark results, we rethink the scalability issue of GCNs from the perspective of ensembling and then present an ensembling-based trainer scheme (EnGCN) that solely needs to train a couple of simple MLPs to achieve new SOTA on multi-scale large datasets. We hope our study on benchmarking and rethinking to help lay a solid, practical, and systematic foundation for the scalable GCN community and provide researchers with broader and deeper insights into large-scale graph training.

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
