# A1 More details of Formulations

## A1.1 Representative Subgraph Sampling Schemes

⋆ **Node Sampler** [25, 24]: $\mathbb{P}(u) = ||\widetilde{\mathbf{A}}_{:,u}||^2$, where all nodes are sampled independently based on the normalized distribution of $\mathbb{P}$. This sampling strategy is logically equivalent to layer-wise sampling [25].

⋆ **Edge Sampler** [24]: $\mathbb{P}(u,v) = \frac{1}{deg(u)} + \frac{1}{deg(v)}$, where all edges are sampled independently based the edge distribution above. In our implementation, we utilize the sampled nodes (once contained in the sampled edges) to induce the subgraph as input, which should include more edges to help boost the performance.

⋆ **Random Walk Sampler** [61, 24]: Here, we first sample a subset of root nodes uniformly, based on which we perform a random walk at a certain length to obtain the subgraph as a batch.

⋆ **Graph Partitioner** [23, 39]: We first partition the entire graph into clusters with graph clustering algorithms and then select multiple clusters to form a batch.

# A2 Additional Experiment Results

## A2.1 Additional Hyperparameter Searching Results

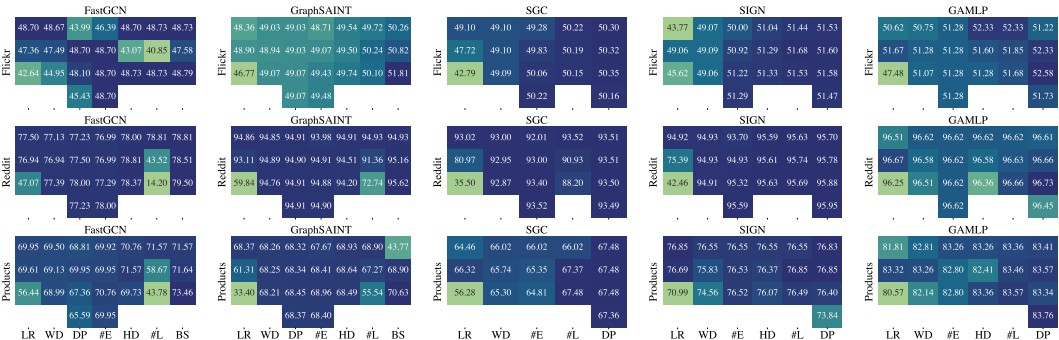

Figure A5: The greedy hyperparameter searching results for other methods.

Table A5: The searched optimal hyperparameters for all tested methods

| Category | Methods | Datasets | | |
| --- | --- | --- | --- | --- |
| | | Flickr | Reddit | ogbn-products |
| Sampling | GraphSAGE [3] | LR: 0.0001, WD: 0.0001, DP: 0.5, EP: 50, HD: 512, #L: 4, BS: 1000 | LR: 0.0001, WD: 0.0 DP: 0.2, EP: 50, HD: 512, #L: 4, BS: 1000 | LR: 0.001, WD: 0.0 DP: 0.5, EP: 50, HD: 512, #L: 4, BS: 1000 |
| | FastGCN [25] | LR: 0.001, WD: 0.0002, DP: 0.1, EP: 50, HD: 512, #L: 2, BS: 5000 | LR: 0.01, WD: 0.0 DP: 0.5, EP: 50, HD: 256, #L: 2, BS: 5000 | LR: 0.01, WD: 0.0 DP: 0.2, EP: 50, HD: 256, #L: 2, BS: 5000 |
| | LADIES [26] | LR: 0.001, WD: 0.0002, DP: 0.1, EP: 50, HD: 512, #L: 2, BS: 5000 | LR: 0.01, WD: 0.0001 DP: 0.2, EP: 50, HD: 256, #L: 2, BS: 5000 | LR: 0.01, WD: 0.0 DP: 0.2, EP: 30, HD: 256, #L: 2, BS: 5000 |
| | ClusterGCN [23] | LR: 0.001, WD: 0.0002, DP: 0.2, EP: 30, HD: 256, #L: 2, BS: 5000 | LR: 0.0001, WD: 0.0 DP: 0.5, EP: 50, HD: 256, #L: 4, BS: 2000 | LR: 0.001, WD: 0.0001 DP: 0.2, EP: 40, HD: 128, #L: 4, BS: 2000 |
| | GraphSAINT [24] | LR: 0.001, WD: 0.0004, DP: 0.2, EP: 50, HD: 512, #L: 4, BS: 5000 | LR: 0.01, WD: 0.0002 DP: 0.7, EP: 30, HD: 128, #L: 2, BS: 5000 | LR: 0.01, WD: 0.0 DP: 0.2, EP: 40, HD: 128, #L: 2, BS: 5000 |
| Decoupling | SGC [27] | LR: 0.01, WD: 0.0002, EP: 100, #L:2, DP: 0.5 | LR: 0.01, WD: 0.0001, EP: 50, #L:2, DP: 0.1 | LR: 0.001, WD: 0.0001, EP: 500, #L:8, DP: 0.1 |
| | SIGN [28] | LR: 0.001, WD: 0.0002, EP: 100, HD:256, #L:4, DP: 0.2 | LR: 0.01, WD: 0.0002, EP: 50, HD: 512, #L:8, DP: 0.7 | LR: 0.01, WD: 0.0001, EP: 500, HD:256, #L:4, DP: 0.2 |
| | SAGN [29] | LR: 0.01, WD: 0.0001, EP: 20, HD:64, #L:4, DP: 0.7 | LR: 0.001, WD: 0.0002, EP: 50, HD: 256, #L:2, DP: 0.5 | LR: 0.001, WD: 0.0, EP: 500, HD:512, #L:4, DP: 0.5 |
| | GAMLP [30] | LR: 0.001, WD: 0.0002, EP: 20, HD:64, #L:2, DP: 0.5 | LR: 0.001, WD: 0.0001, EP: 30, HD: 128, #L:2, DP: 0.5 | LR: 0.001, WD: 0.0002, EP: 500, HD:768, #L:8, DP: 0.7 |
| | LP [38, 40] | DT: residual, #Prop: 20, AR: 0.9, Adj: $D^{-1/2}AD^{-1/2}$, AS: True, #ML:2 | DT: residual, #Prop: 50, AR: 0.9, Adj: $D^{-1}A$, AS: True, #ML:2 | DT: residual, #Prop: 20, AR: 0.9, Adj: $D^{-1}A$, AS: True, #ML:3 |

## A2.2 A Joint Comparison of Effectiveness and Efficiency

To further facilitate a comprehensive understanding of the benchmark results, we provide an illustration in Figure A6 to jointly compare the effectiveness and efficiency of the methods. An empirical summary could be found in Section 5.1.

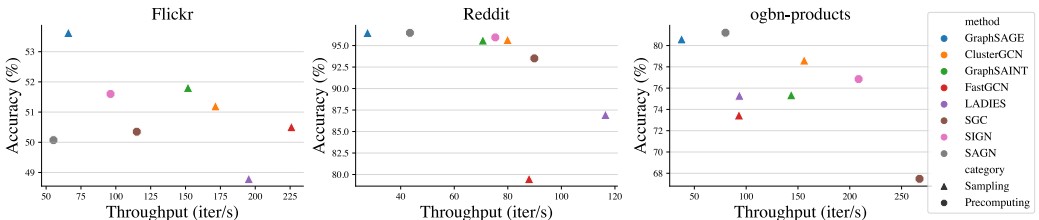

Figure A6: The joint comparison of effectiveness (accuracy) and efficiency (throughput) for sampling-based and precomputing-based methods.

## A2.3 The Hyperparameter Settings for EnGCN

The searched HPs for EnGCN includes learning rate (0.01, 0.001, 0.0001), weight decay (0, 1e-5, 1e-4), dropout (0.2, 0.5, 0.7), epochs (30, 50, 70), hidden dimension (128, 256, 512), batch size (5000, 10000), batch norm (True, False), self learning threshold ($\alpha$=0.8, 0.9, 0.95), and number of layers (4, 5, 8). The searching results are shown in Figure A7. The searched HPs that produce the reported results on Flickr, Reddit, and ogbn-products are shown in Table A6.

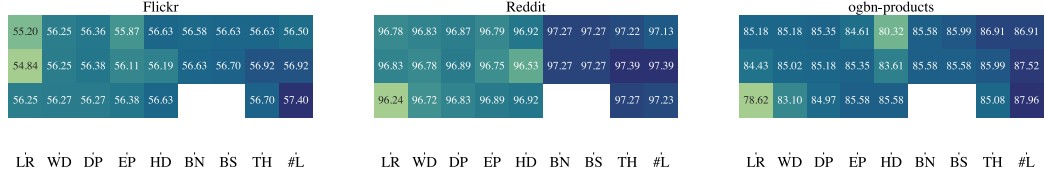

Figure A7: The hyperparameter searching results of EnGCN.

Table A6: The searched optimal hyperparameters for EnGCN on Flickr, Reddit, and ogbn-products

| Datasets | Searched HPs |
|---|---|
| Flickr | lr 0.0001 weight decay 0.0001 dropout 0.2 epoch 70 hidden dimension 256 number of layers 4 batch size 10000 $\alpha$ 0.9 |
| Reddit | lr 0.001 weight decay 0 dropout 0.2 epochs 70 hidden dimension 512 number of layers 4 batch size 5000 $\alpha$ 0.95 |
| ogbn-products | lr 0.01 weight decay 0 dropout 0.2 epochs 70 hidden dimension 512 number of layers 8 batch size 10000 $\alpha$ 0.8 |

## A2.4 Ablation Study for EnGCN

**Ablating Ensembling.** Here we provide an ablation study to confirm the effectiveness of ensembling (inference with majority voting). For ablated models, we directly use the ones after $l$-hop training, $0 \leq l \leq 3$. The experiment results are shown in Table A7. Notably, with majority voting, the performance is boosted by a large margin on Flickr and also has noticeable improvement on Reddit and ogbn-products. Besides, we find that the test accuracy of EnGCN after $l$-hop training keeps increasing as $l$ grows. This phenomenon is consistent with the empirical results in AdaGCN [56].

Table A7: The test accuracy (%) for ablated EnGCNs.

| Category | Flickr | Reddit | ogbn-products |
|---|---|---|---|
| EnGCN after 0-hop training | 46.11±0.14 | 74.51±0.09 | 61.97±0.08 |
| EnGCN after 1-hop training | 46.26±0.17 | 94.26±0.05 | 83.48±0.13 |
| EnGCN after 2-hop training | 50.00±0.49 | 95.23±0.03 | 87.69±0.06 |
| EnGCN after 3-hop training | 50.56±0.80 | 95.28±0.04 | 87.80±0.33 |
| EnGCN with majority voting | **56.43±0.21** | **97.14±0.03** | **87.99±0.04** |

**Ablating SLE.** Here we provide another simple ablation study to confirm the contribution of SLE to EnGCN. From another perspective, the results further demonstrate the importance of label propagation in graph representation learning, especially for large-scale graphs.

Table A8: The accuracy (%) of ablating SLE from EnGCN

| Methods | Flickr | Reddit | ogbn-products |
|---|---|---|---|
| EnGCN w.o. SLE | $50.22 \pm 0.30$ | $96.92 \pm 0.10$ | $76.35 \pm 0.06$ |
| EnGCN | $\mathbf{56.43 \pm 0.21}$ | $\mathbf{97.14 \pm 0.03}$ | $\mathbf{87.99 \pm 0.04}$ |

## A3 Additional Implementation Details

### A3.1 Access and Statistics of Benchmark Datasets

All datasets we used could be accessed through the APIs provided py PyTorch Geometric[3] [62]. The statistics of Flickr, Reddit, and ogbn-products are provided as follows.

Table A9: The statistics of Flickr, Reddit, and ogbn-products

| Dataset | Nodes | Edges | Classes | splitting (Train/Validation/Test) | Task |
|---|---|---|---|---|---|
| Flickr | 89,250 | 899,756 | 7 | 0.50 / 0.25 / 0.25 | Multi-Class Classification |
| Reddit | 232,965 | 11,606,919 | 41 | 0.66 / 0.10 / 0.24 | Multi-Class Classification |
| ogbn-products | 2,449,029 | 61,859,140 | 47 | 0.10 / 0.02 / 0.88 | Multi-Class Classification |

### A3.2 Implementation details of testing GPU memory and throughput

Here we provide the details of implementation and hyperparameters for the throughput and memory usage experiments. Regarding the implementation, we evaluate the hardware throughput based on Chen et.al. [63]. For the activation memory, we measure it based on `torch.cuda.memory_allocated`.

Regarding the hyperparameter setting in the throughput and memory usage measurement, we set the hidden dimension to 128 across different models and datasets. We control the number of nodes whose embedding requires gradients roughly equal to 5,000 across different models and datasets. Thus, our method is fair in the sense that we control the number of active nodes per batch in the same for different methods. We note that for graph-wise sampling-based methods (e.g., ClusterGCN, GraphSAINT), the number of nodes whose embedding requires gradients equals the number of nodes retained in the GPU memory. However, for other sampling-based methods (e.g., GraphSAGE, FastGCN), they need to gather the neighbor embeddings to update the node embedding in the current batch. These embeddings of nodes that are outside the current batch do not require gradients. We also want to clarify that the hyperparameter "batch_size" in our script has a different meaning for different methods. For example, for precomputing methods, a 5,000 "batch_size" means each mini-batch contains 5,000 input samples (i.e., nodes). For GraphSAINT, "batch_size" means the number of roots in the random walk sampler. Thus, the number of nodes in each mini-batch roughly contains "batch_size" $\times$ "walk_length".

## A4 Intended Use

The license of our repository is MIT license. For more information, please refer to `https://github.com/VITA-Group/Large_Scale_GCN_Benchmarking/blob/main/LICENSE`. Our benchmark is for researchers and scientists in graph mining and data science community to propose innovative methods, especially for large-scale graph training. We implement a number of representative scalable GNN models, provide several abstract classes for further inheriting, and define a unified training process for a fair comparison. In our code base, we implement two abstract classes for *sampling-based* and *precomputing-based* methods based on our unified formulations in Section 2 , respectively. One could build up his/her new sampling-based or precomputing-based GNN models upon the code base by solely overwriting a few specific functions. For detailed usage including installation, reproduction, etc., please refer to our documentation in the repository.

---

[3]`https://github.com/pyg-team/pytorch_geometric`