# OpenReview forum: "A Comprehensive Study on Large-Scale Graph Training: Benchmarking and Rethinking"
_NeurIPS.cc/2022/Track/Datasets_and_Benchmarks — NeurIPS 2022 Datasets and Benchmarks _

### Official Review · Reviewer_oPP2 · 2022-07-19

**Rating:** 6
**Confidence:** 4
**Correctness:** Yes
**Clarity:** Yes

**Strengths:**

(+) Large-scale GNNs are a trending and critical problem.
(+) The authors have compared various GNN methods with a unified evaluation protocol, including how to tune hyperparameters.
(+) The proposed EnGCN shows impressive performance.
(+) The proposed benchmark is publicly available at Github.


**Weaknesses:**

Limitations and suggestions/questions:
(-) The adopted datasets are somewhat limited. I acknowledge that two datasets (Flickr and Reddit) are social networks and ogbn-products is a co-purchasing graph, but more diverse datasets (e.g., citation and biochemical) and tasks (e.g., link prediction or graph classification, besides node classification) could provide a more comprehensive picture for large-scale GNNs.
(-) Since this paper aims to provide "a fair and consistent benchmark", another essential part besides showing results for the existing methods is to allow new methods to be compared using this benchmark. Therefore, I think some discussions regarding this aspect should be provided, e.g., how others should use this benchmark and whether there are limitations.
(-) Besides separately showing accuracy and efficiency (in Table 1 and 3, respectively), I think it would make the tradeoffs more straightforward if these metrics could be shown jointly, e.g., in a Pareto surface.
(-) I wonder how the exact implementation would affect the efficiency of different methods. For example, it seems that the authors have adopted PyG, and would another library, e.g., DGL, change the results?


**Additional Feedback:**

See comments above

**Documentation:**

The reproducibility seems fine (though I didn't run the code), but the documentation (or instructions on how users should use the benchmark) could be enhanced.

**Relation To Prior Work:**

Yes

**Summary And Contributions:**

This paper proposes to study GNNs for large-scale graphs. Specifically, the paper systemically compares sampling-based approaches (including node-wide, layer-wise, and subgraph-wise) and decoupling-based approaches (including preprocessing and post-processing) on three datasets. Besides reporting several observations for the existing methods, this paper also proposes EnGCN, an ensembling-based approach that achieves SOTA performance.

---

> ### Author Response · Authors · 2022-08-17
> **Response to reviewer oPP2**
>
> Dear reviewer oPP2:
>
> We appreciate your careful reviews. We have addressed your comments as follows.
>
> **Comment #1:**
> The adopted datasets are somewhat limited. I acknowledge that two datasets (Flickr and Reddit) are social networks and ogbn-products is a co-purchasing graph, but more diverse datasets (e.g., citation and biochemical) and tasks (e.g., link prediction or graph classification,
> besides node classification) could provide a more comprehensive picture for large-scale GNNs.
>
> **Reply:**
> Thanks for the comments. We select the three datasets based on the consideration of various scales, where Flickr has about 80,000 nodes; Reddit has about 230,000 nodes; and ogbn-products has 2,449,029 nodes. Three of them could sufficiently provide a relatively comprehensive comparison and study. We’ll keep adding more types of graphs and more representative datasets, e.g. ogbn-arxiv, in the following months along the continual improvement of this work/paper.
>
> Regarding different tasks, we believe it is not feasible in the sense that the three tasks are disjoint to each other, and the backbones and methods to solve them are also different. Specifically, the scalability issue is also different in the three tasks. For example, in graph
> classification, where the datasets are relatively a set of graphs, the scalability issue is not as severe as node classification since we could easily split the set of graphs into batches.
>
> **Comment #2:**
> Since this paper aims to provide "a fair and consistent benchmark", another essential part besides showing results for the existing methods is to allow new methods to be compared using this benchmark. Therefore, I think some discussions regarding this aspect should be provided, e.g., how others should use this benchmark and whether there are limitations.
>
> **Reply:**
> Thanks for the comments. We have added an instruction about intended use in Appendix A4. For your reference, we copy it here.
>
> The license of our repository is MIT license. For more information, please refer to MIT License - VITA-Group/Deep_GCN_Benchmarking - GitHub. Our benchmark is for researchers and scientists in the graph mining and data science community to propose innovative methods especially for large-scale graph training. We implement a number of representative scalable GNN models, provide several abstract classes for further inheriting, and define a unified training process for fair comparison. In our code base, we implement two abstract classes for sampling-based and precomputing-based methods based on our unified formulations in Section 2 , respectively. One could build up his/her new sampling-based or precomputing-based GNN models upon the code base by solely overwriting a few specific functions. For the detailed usage including installation, reproduction, etc., please refer to our documentation in the repository.
>
>
> **Comment #3:**
> Besides separately showing accuracy and efficiency (in Table 1 and 3, respectively), I think it would make the tradeoffs more straightforward if these metrics could be shown jointly, e.g., in a Pareto surface.
>
> **Reply:**
> Thanks for the suggestions. We have added a new figure (Fig. A6) in Appendix 2.2 to jointly compare the throughput and accuracy of tested methods.
>
> **Comment #4:**
> I wonder how the exact implementation would affect the efficiency of different methods. For example, it seems that the authors have adopted PyG, and would another library, e.g., DGL, change the results?
>
>
> **Reply:**
> Thanks. we would like to explain it in terms of *effectiveness* and *efficiency*. Firstly, the performance (effectiveness) should be consistent with the other commonly-used graph learning libraries, e.g. DGL. This is because the performance is determined by the algorithms and also both PyG and DGL have provided a reproducible framework that we can build upon. We are thankful for the team's valuable efforts. Secondly, the corresponding efficiency of different implementation frameworks, such as the training time, CPU and GPU memory usage, is supposed to be different, because these metrics are highly related to the specific implementations. For example, the storage type of adjacent matrices would affect the memory usage and the "*spmm*" operation makes a non-trivial influence on the inference time, etc.
>
>
> Best,
>
> Authors

---

> > ### Comment · Reviewer_oPP2 · 2022-08-25
> > **Response to Rebuttal**
> >
> > I have read the rebuttal and thank the authors for addressing my concerns. I will keep the current score and encourage the authors to add these discussions in the paper.

---

### Official Review · Reviewer_vowK · 2022-07-23
**This paper is comprehensive and meaningful. This paper introduces the large-scale graph training vey well.**

**Rating:** 8
**Confidence:** 4
**Correctness:** I think the claims in the paper are c…
**Clarity:** The writing of the paper is good. The…

**Strengths:**

1. The provided benchmark is comprehensive and looks fair.
2. The evaluation matrics are helpful and easy to obtain.
3. The experimental results are comprehensive.
4. The study is solid and helpful.

**Weaknesses:**

I think the consistency and robustness should also be considered in the evaluation.

**Additional Feedback:**

No.

**Documentation:**

I think the experiments discussed in the paper could be reproduced.

**Ethics:**

No.

**Relation To Prior Work:**

The paper clearly discusses its relation to prior work. It's obviously a new work.

**Summary And Contributions:**

The scalability issue of graph convolutional networks has been being a notoriously challenging
research problem. This paper establish a benchmark for large-scale graph training w.r.t effectiveness and efficiency. And a unified formulation is given for dozens of works and further assess on the basis of accuracy, memory usage, throughput, and convergence. And rethinking of the scalability of GCNs is provided from the perspective of ensembling and ensembling-based trainer scheme.

---

> ### Author Response · Authors · 2022-08-17
> **Response to reviewer vowK**
>
> Dear reviewer vowK:
>
> We really appreciate your positive assessment of our work. W.r.t. your comment, “I think the consistency and robustness should also be considered in the evaluation”, we have further polished our manuscript and added more empirical analysis and illustration to make it comprehensive and consistent.
>
> Best,
>
> Authors

---

### Official Review · Reviewer_W47P · 2022-07-27
**An interesting benchmark paper but more experiments are needed**

**Rating:** 6
**Confidence:** 4
**Correctness:** Good.
**Clarity:** Good.

**Strengths:**

The problem is interesting and important. The scalable GNNs have been widely studied in the machine learning community. It could be beneficial to benchmark all the scalable GNNs and summarize the pros and cons.

Insightful observations are provided. The authors claimed various observations with respect to effectiveness and efficiency. Some of them are interesting findings.

The authors propose a new GNN to solve the constraints of sampling-based methods and precomputing-based methods.


**Weaknesses:**

More experiments can be done to validate the effectiveness of EnGCN. For example, what is the parameter sensitivity of EnGCN, considering the reported performance strongly relies on hyper-parameter search? What is the performance of EnGCN on other public benchmark datasets (e.g., ogbn-proteins, ogbn-arxiv), besides Flickr, Reddit, and ogbn-products?

Ablation studies are needed to verify the effectiveness of each component in EnGCN, e.g., pre-processing, post-processing, and majority voting.


**Additional Feedback:**

Please see comments above.

**Documentation:**

Good.

**Ethics:**

N/A.

**Relation To Prior Work:**

Good.

**Summary And Contributions:**

The authors benchmark large-scale graph training for both sampling-based methods and decoupling-based methods. The authors also compare the efficiency and analyze the pros and cons of different GNNs. Besides, the authors present EnGCN, a new ensembling training manner.

---

> ### Author Response · Authors · 2022-08-17
> **Response to reviewer W47P**
>
> Dear reviewer W47P:
>
> We are thankful for your recognition of our research motivation and contributions. We carefully considered the suggestions of conducting more experiments. In this feedback, we provide several additional empirical results to hopefully address your concerns to some extent.
>
> **Comment #1:**
> More experiments can be done to validate the effectiveness of EnGCN. For example, what is the parameter sensitivity of EnGCN, considering the reported performance strongly relies on hyper-parameter search? What is the performance of EnGCN on other public benchmark datasets (e.g., ogbn-proteins, ogbn-arxiv), besides Flickr, Reddit, and ogbn-products?
>
> **Reply:**
> Thanks for the suggestions.
>
> For the HP parameter searching results of EnGCN, we have included it in Appendix A2.3 (Figure A7). As shown, even with default HP settings, EnGCN is capable of achieving ideal performance. For searched parameters, EnGCN is relatively more sensitive to number of layers (#L), where the accuracy grows as #L increases.
>
> For more benchmark datasets, we will conduct the experiments in our camera-ready version if we are fortunate to be accepted.
>
> **Comment #2:**
> Ablation studies are needed to verify the effectiveness of each component in EnGCN, e.g., pre-processing, post-processing, and majority voting.
>
> **Reply:**
> Thanks for the suggestions. In the original version, we provided an ablation study of majority voting. In the revised version, following the suggestions, we add an additional ablation study of SLE for completeness in Section A2.3. For your reference, we copy it here.
>
> **Ablating Ensembling**. Here we provide an ablation study to confirm the effectiveness of ensembling (inference with majority voting). For ablated models, we directly use the ones after l-hop training, 0 <= l <= 3. The experiment results are shown in the table below. Notably, with majority voting, the performance is boosted by a large margin on Flickr and also has noticeable improvement on Reddit and ogbn-products. Besides, we find that the test accuracy of EnGCN after l-hop training keeps increasing as l grows. This phenomenon is consistent with the empirical results in AdaGCN.
>
> | Category                   | Flickr         | Reddit         | ogbn-products  |
> | -------------------------- | -------------- | -------------- | -------------- |
> | EnGCN after 0-hop training | 46.11±0.14     | 74.51±0.09     | 61.97±0.08     |
> | EnGCN after 1-hop training | 46.26±0.17     | 94.26±0.05     | 83.48±0.13     |
> | EnGCN after 2-hop training | 50.00±0.49     | 95.23±0.03     | 87.69±0.06     |
> | EnGCN after 3-hop training | 50.56±0.80     | 95.28±0.04     | 87.80±0.33     |
> | EnGCN with majority voting | **56.43±0.21** | **97.14±0.03** | **87.99±0.04** |
>
> **Ablating SLE.** Here we provide another simple ablation study to confirm the contribution of SLE to EnGCN. From another perspective, the results further demonstrate the importance of label propagation in graph representation learning, especially for large-scale graphs.
>
>
> | Methods        | Flickr       | Reddit       | ogbn-products |
> | -------------- | ------------ | ------------ | ------------- |
> | EnGCN w.o. SLE | 50.22 ± 0.30 | 96.92 ± 0.10 | 76.35 ± 0.06  |
> | EnGCN          | 56.43 ± 0.21 | 97.14 ± 0.03 | 87.99 ± 0.04  |
>
>
> Best,
>
> Authors

---

### Official Review · Reviewer_W3bB · 2022-07-28
**Benchmarking of standard models on a few common datasets, adding new strong results based on an ensemble method.**

**Rating:** 7
**Confidence:** 3

**Strengths:**

The main strengths of this work are 1) the clarity of the taxonomy of the benchmarked methods 2) the detailed observational results, of use as a reference by the community, and 3) the strong performance of the proposed method, EnGCN.

On 3) The performance of the EnGCN method on ogbn-products is impressive, conditional on replicability and comparable computational cost (see related comment).

**Weaknesses:**

Weaknesses are limited, and mostly do not compromise the overall value of the work - 1) hyperparameter search setup is simple, potentially biased, and 2) presentation of empirical efficiency results could be improved, and such results for EnGCN are missing.

**Additional Feedback:**

N/A

**Clarity:**

Generally the paper is well organized and clearly written. A close pass for typos will be helpful.

On Strength 1) - Notational conventions are simple but adequate to describe the differences between the methods as relevant to their benchmark analysis. The hyperparameter search space and results grids are clearly presented in Table/Fig 1, and the best settings are collated well in appendix A5.

On Weakness 2) - The empirical efficiency analysis is presented using "Throughput" in (iterations/s) which is not clearly defined. Stated differently, it is hard to understand the comparison, or relative point between those numbers in Table 3. Potentially a "pareto-frontier" curve based on different methods training time and performance could be generated, though technically one can extract this information from Table 2.

Related to the above, the Appendix section A2.3, Figure A4 and Table 7, may have some typos in how the accompanying prose references them. I was a bit confused.

Selected typos:
- I suggest replacing the "sweet point" motif with "sweet spot" which is the more common english phrase.
- Pg 4, Post-processing section, Zeros method definition, script T missing the "train" subscript
- I believe Table 1 should not say "sampling" methods only
- Obs 5, remove "Counter-intuitively" or explain why it is framed this way?



**Correctness:**

Overall the work is sound and likely presents correct results.

On Strength 2) - Obs 2 matches the standard takeaway in the literature, and Obs 3 is plausible, their hparam search seems fruitful. Obs 7 is a very practical summary takeaway, backed up by their data, the sort of thing a practitioner will appreciate.

On Weakness 1) - It is possible that a more robust sampling strategy over hyperparameters should have been employed. Minimally it might be useful to ablate over the order (of the different parameter columns in Fig 1) in which the greedy optimization is performed. Certain parameters like batch size and depth are interrelated with respect to optimization behavior.

Related to the latter comment, while benchmarking on a commodity GPU like a 3090 is valuable since it is accessible to most researchers, it does limits the hparam values that can be explored, ie. the OOM errors. Some extra runs on a larger card would be nice to exhaust the batch and depth degrees of freedom.

On Weakness 2) - It would be important to understand the relative efficiency or cost of the proposed method EnGCN to the others given that the work presents it as SOTA, however efficiency results are not included (unless I missed something?).

**Documentation:**

The main work plus appendix describes the analysis well, and the codebase is provided with adequate setup documentation. Assumption is that the results are reproducible given the standardized source of data through PyG.

**Relation To Prior Work:**

This work does not present any single component or analysis that is particularly novel standing alone (except the new method) however, the consolidation of the different theoretical and empirical analyses is of significant value as a single point of reference.

As the authors state, EnGCN is relatively similar to AdaGCN, however, they appear to perform at quite different levels.

**Summary And Contributions:**

In this work, authors perform a benchmark analysis by applying a selection of models to three common datasets. They provide a basic theoretical description of the two major groups of algorithms, sampling and decoupling, and compare the pros and cons of the various approaches. Their empirical evaluation provides evidence for a series of practically useful observations that align with theoretical claims. Finally they propose a semi-novel ensemble method that is competitive on the datasets considered, from a performance perspective.

---

> ### Author Response · Authors · 2022-08-17
> **Response to reviewer W3bB**
>
> Dear reviewer W3bB,
>
> We really appreciate your recognition of our work and thanks for the detailed and valuable comments. We carefully revised our paper based on the suggestions and hopefully it could address all your concerns.
>
> **Comment #1:**
> It is possible that a more robust sampling strategy over hyperparameters should have been employed. Minimally it might be useful to ablate over the order (of the different parameter columns in Fig 1) in which the greedy optimization is performed. Certain parameters like batch size and depth are interrelated with respect to the optimization behavior.
>
> **Reply:**
> We are thankful for the suggestions about ablating the HP orders. We will include a case study in our updated version.
>
> **Comment #2:**
> Related to the latter comment, while benchmarking on a commodity GPU like a 3090 is valuable since it is accessible to most researchers, it does limits the hparam values that can be explored, ie. the OOM errors. Some extra runs on a larger card would be nice to exhaust the batch and depth degrees of freedom.
>
> **Reply:**
> Thanks for the suggestion. We reran the "OOM" case on another GPU with large memory and got 80.56% accuracy. This is also reported in our revised version as a footnote.
>
> **Comment #3:**
> It would be important to understand the relative efficiency or cost of the proposed method EnGCN to the others given that the work presents it as SOTA, however efficiency results are not included (unless I missed something?).
>
> **Reply:**
> Thanks for the suggestions. We have added a detailed effectiveness evaluation of EnGCN in our revised version (Section 5.5 and Section 5.6) to empirically analyze the CPU memory consumption, GPU training efficiency, and convergence landscape (which is placed in the appendix originally).
>
> **Comment #4:**
> The empirical efficiency analysis is presented using "Throughput" in (iterations/s) which is not clearly defined. Stated differently, it is hard to understand the comparison, or relative point between those numbers in Table 3. Potentially a "pareto-frontier" curve based on different methods training time and performance could be generated, though technically one can extract this information from Table 2.
>
> **Reply:**
> Many thanks for the constructive comments. Regarding the definition of throughputs, we add an explanation in our updated manuscript: Here “Throughput” measures how many times can we complete the training steps within a second. In addition, similar to the suggestions of "pareto-frontier" curve, for a joint comparison of the effectiveness and efficiency of the benchmarked methods, we provide a scatter plot where the x-axis denotes the throughput and the y-axis denotes the accuracy. We include it in our updated appendix (Figure A6).
>
>
> **Comment #5:**
> A close pass for typos will be helpful. The Appendix section A2.3, Figure A4 and Table 7, may have some typos in how the accompanying prose references them. I was a bit confused.
>
> **Reply:**
> Thanks very much for the detailed comments. For the selected typos, we have closely revised the manuscript and fixed the grammar errors mentioned. For the hyperlinking, we have checked it and it turns out to be correct. The misunderstanding is due to the close position between Table A7 and Figure A4. We have rearranged their positions to avoid misunderstandings.
>
> Best,
>
> Authors.

---

### Official Review · Reviewer_s5iV · 2022-07-28

**Rating:** 6
**Confidence:** 4
**Correctness:** Claims are correct and evaluations ar…

**Strengths:**

This paper has a good summarization and review of previous works.

This paper provides extensive experimental benchmark results of performance scores and time & memory overhead.

This paper provides interesting observations and intuitions from the benchmark.

This paper proposes EnGCN which achieves SOTA performances.

**Weaknesses:**

It seems that EnGCN is a variant of SIGN with three additional components: 1. label Y as feature, 2. self-training, 3. voting (boosting). The performance of EnGCN is quite strong but the authors fail to ablate on those components to deliver intuitions behind the designs. Also, EnGCN will share the “Cons” of the preprocessing method, which is consuming a large volume of CPU memory when training on large-scale graphs, and the authors do not cover the efficiency of the proposed method in the paper. Lastly, the authors need to provide a more detailed description of the methodology. Currently, the writing is vague and hard to understand.

**Additional Feedback:**

No further comments.

**Clarity:**

At large, this paper is clearly written and easy to follow, but the writing of the methodology of EnGCN needs further polishing.

Typos: page 7, GINAT→GIANT, Lable→Label

Section 4.2: Does the speed meassure of decoupling-based methods count the operations on CPU? How are the hyperparameters chosen to ensure a fair efficiency comparison?

**Documentation:**

The paper provides open-source implementation with needed instructions for reproducibility. The paper only uses public datasets.

**Ethics:**

No ethical concerns.

**Relation To Prior Work:**

The authors may also consider memory-based scalable algorithms for reference.

GNNAutoScale: Scalable and Expressive Graph Neural Networks via Historical Embeddings, ICML2021

VQ-GNN: A Universal Framework to Scale up Graph Neural Networks using Vector Quantization, NeurIPS2021

**Summary And Contributions:**

This paper studies the important question of the scalability of GNNs on the large-scale node classification task. In the first half, the paper clearly categorizes literature into sampling- and decoupling-based methods and provides an extensive benchmark in terms of performance and efficiency. In the second half, the paper leverages the intuition learned from the first half and proposes their ensemble method (EnGCN) that registers strong performances. Both parts are of valuable contributions.

---

> ### Author Response · Authors · 2022-08-17
> **Response to reviewer s5iV**
>
> Dear reviewer s5iV:
>
> We appreciate your recognition of our benchmark and technique contributions. We have addressed all your concerns as follows.
>
> **Comment #1:**
> The performance of EnGCN is quite strong but the authors fail to ablate on those components to deliver intuitions behind the design.
>
> **Reply:**
> We appreciate your recognition of the empirical performance of our proposed method, and we believe this concern can be addressed with more detailed clarification of EnGCN.
>
> 1. Technically, EnGCN is not a variant of SIGN since it is a different training scheme from *precomputing* as we stated in our motivation (section 5.2). EnGCN is primarily proposed to address the large CPU occupation issue of the precomputing-based methods, where a number of large precomputed feature matrices are *simultaneously* stored in the CPU for further use (e.g. the concatenation in SIGN and the attention mechanism in SAGN).
>
> 2. As of the comments of "three additional components", we introduce our intuitions as follows: (i) "*label Y as feature*" and "*self-training*" constitutes the *SLE*, which we organically combine to boost the prediction performance. To further confirm the importance of SLE in EnGCN, we provide a simple ablation study in Table, which we also include in our revised version. (ii) "Voting" is crucial for EnGCN as it is the essential thought of Ensembling-based methods. We also include an ablation study in our original paper version (Appendix A2.4) to validate it.
>
>
>     | Methods        | Flickr         | Reddit         | Ogbn-products  |
>     |----------------|----------------|----------------|----------------|
>     | EnGCN w.o. SLE | 50.22 ± 0.30 % | 96.92 ± 0.10 % | 76.35 ± 0.06 % |
>     | EnGCN          | 56.43 ± 0.21%  | 97.14 ± 0.03%  | 87.99 ± 0.04%  |
>
> **Comment #2:**
> EnGCN "shares" the cons of the preprocessing methods, which is consuming a large volume of CPU memory when training on large-scale graphs, and the authors do not cover the efficiency of the proposed method in the paper.
>
> **Reply:**
> We appreciate your constructive suggestions about the efficiency of EnGCN. Regarding the statement that "EnGCN shares the cons of the preprocessing methods", we would like to clarify some potential misunderstandings:
>
> Precomputing-based methods, such as SIGN, need to store all of the intermediate feature matrix as $\bar{\mathbf{X}} = \begin{bmatrix}\mathbf{X} & \mathbf{A}\mathbf{X} & \cdots & \mathbf{A}^K\mathbf{X}\end{bmatrix}$. In contrast, EnGCN does not store any intermediate message-passed feature matrix, i.e., $\mathbf{A}\mathbf{X}$, $\mathbf{A}^2\mathbf{X}$, etc.* Specifically, in EnGCN, For the $i$-th layer, we first compute $\mathbf{X}^{(l)}=\mathbf{A}\mathbf{X}^{(l-1)}$. Then
> $\mathbf{X}^{(l-1)}$ will be overwritten and the memory will be released. Only $\mathbf{X}^{(l)}$ will be used to train the predictor, e.g. a MLP, till convergence. As a result, EnGCN occupies significantly less memory compared to pre-computing based methods. For example, according to our experiments, the $K$ in SIGN is often set to 8 to obtain optimal performance. In this case, EnGCN has $8\times$ less CPU memory consumption compared to SIGN.
>
> For further empirical confirmation, we provide the efficiency evaluation of EnGCN in terms of CPU memory usage in Figure 4 (Page 10), where the x-axis denotes the models' number of layers while the y-axis records their allocated memories that are reported by "*aten::empty*" of PyTorch. As shown in Figure 4, the precomputing-based methods, SIGN and SAGN, suffer from expensive CPU memory consumption as the model depth increases. For sampling-based methods, since there is no need to pre-store a large number of feature matrices, the memory consumption increases much more smoothly. For EnGCN, as no precomputing is required, the CPU memory consumption is considerably reduced in comparison with SIGN and SAGN, which intuitively validates the CPU memory efficiency of EnGCN.
>
> **Comment #3:**
> Typos: page 7, GINAT $\rightarrow$ GIANT, Lable $\rightarrow$ Label
>
> **Reply:**
> Thanks for your comments. We have fixed the typos mentioned in our updated version.
>
> **Comment #4:**
> Does the speed measure of decoupling-based methods count the operations on CPU?
>
> **Reply:**
> No. We only compare the GPU training speed among decoupling-based and sampling-based methods for a fair comparison.
>
> **Comment #5:**
> How are the hyperparameters chosen to ensure a fair efficiency comparison.
>
> **Reply:**
> We have provided the corresponding hyperparameter settings and a detailed discussion in Appendix A3.2 in our original version. Hopefully, it could address your concerns.
>
> **Comment #6:**
> Add the memory-based scalable algorithms for reference
>
> **Reply:**
> Many thanks for the constructive suggestions to help enrich our survey. We include the related works in our revised version at section 2.3.
>
> If you find our updates helped address your concerns, we sincerely hope you could upscore your evaluation for our work.
>
> Best,
>
> Authors

---

### Official Review · Reviewer_kf8i · 2022-07-29
**A survey with empirical analysis, and an ensemble-based training scheme.**

**Rating:** 6
**Confidence:** 4
**Clarity:** Yes, the paper is sufficiently well-w…

**Strengths:**

This paper not only involves major representative methods on training scalable GNN on large-scale graph, but also categorize these methods into two branches: Sampling-based and Decoupling-based methods. This paper further categorizes the decoupling-based methods into two varieties (the Pre-processing methods and Post-processing methods). To the best of my knowledge, there are surveys and books focus on sampling-based methods, but few surveys concentrate on the decoupling-based methods. This paper provides a fair and consistent evaluation on various representative scalable GNN training methods for large-scale graph. This paper also provides empirical analysis (including empirical Pros. & Cons.) based on the evaluated performance and efficiency. This paper further proposes an ensemble-based training strategy, dubbed as EnGCN, which achieves promising performance compared to various existing scalable GNN training methods.

**Weaknesses:**

Despite the paper provides a relatively "fair and consistent" evaluation (benchmark) for various representative methods, nevertheless, the benchmark has at least two major drawbacks: The HPO strategy is not inevitably insufficient, and the demonstration of evaluation results are not intuitive. Particularly, the linear greedy search strategy (i.e., iteratively searching the optimal value for a HP while fixing the others) is adopted to obtain the best combination of HP, however, the iterative nature possibly not capable to derive near-optimal HP configuration in the search space, besides, the number of trials of HPO for one method is relatively insufficient (less than 30 trials per method on specific dataset). It's understandable that the sufficient HPO process incurs large computational cost. But I am afraid that the empirical results derived by insufficient HPO process are potentially not convincing to support the following empirical analysis, including the empirical Pros.

Cons.
The demonstration of the evaluation results are no sufficiently intuitive and straight-forward, thus it's not easy to show the following empirical analysis. Maybe some statistical properties can be shown to demonstrate the empirical analysis.  Despite the paper introduces an ensemble-based scalable training scheme, however, similar ideas have been proposed (e.g., AdaGCN). Thus the novelty in relatively limited. Nevertheless, the introduced manner has considerable practical capability, that I have already claimed in the Strengths section.

**Additional Feedback:**

Suggestions1: Why not use BO and maybe more computational resources to obtain more convincing evaluation results, by adopting sufficient HPO process.

Suggestion2: Revise the demonstration of the evaluation results. The results should intuitively support the following empirical analysis. Please note: intuitive, straight-forward, and direct.

**Correctness:**

A benchmark is presented in the submission, the evaluation methods and experiment design seems to be correct, but the results are potentially inaccurate, due to the insufficient HPO process.

**Documentation:**

Yes, sufficient details have been presented to support reproducibility, and a URL to GitHub is provided to provide codes and corresponding documentations.

**Ethics:**

No, there are no ethical concerns.

**Relation To Prior Work:**

No, this paper has not discussed the difference with previous related works. Only researchers familiar with related directions are able to recognize the novelty and differences compared with previous works.

**Summary And Contributions:**

This paper provides a relatively comprehensive survey on training scalable GNN for large-scale graphs, and proposes an ensemble-based training strategy dubbed as EnGCN, which achieves promising performance on several large-scale graphs.

---

> ### Author Response · Authors · 2022-08-17
> **Response to reviewer kf8i**
>
> Dear reviewer kf8i:
>
> We really appreciate your agreements on our benchmark contribution for large-scale graph training. We would like to provide more explanations to address the reviewer's concerns.
>
> **Comment #1:**
> Why not use BO and maybe more computational resources to obtain more convincing evaluation results, by adopting sufficient HPO process.
>
> **Reply:**
> We are thankful for your constructive comments. Regarding the hyperparameter (HP) searching strategy, we perform a greedy searching scheme based on two considerations:
>
> 1. First, compared to the grid search and more advanced hyperparameter optimization (HPO) algorithms, e.g. Bayesian Optimization as mentioned, besides the time saving, another reason why we prefer the linear greedy HP search is that it provides us with more insights, which is the prime aim of our benchmark, i.e., providing a comprehensive and systematic
> understanding of previous works. For instance, facilitated with greedy HP search, we could tell the sensitivity of different HPs, which forms our observation 1 in page 5. If we jointly optimize the HPs with specific advanced HPO algorithms, we would somehow lose the interpretability.
>
> 2. Second, the linear greedy HP search is sufficiently effective on finding near-optimal HPs with ideal performance to establish a fair benchmark. Particularly, in comparison with the originally reported results, the greedily searched ones could achieve on-par or even better performance. For example, as we mentioned in Obs. 3, according to our HP search results, GraphSAGE reaches a relatively higher performance than many other sampling-based methods. However, as a commonly-used baseline, the reported performance of GraphSAGE in many comparison results is usually much lower than our search, which does not contribute to a valid and fair comparison.
>
> **Comment #2:**
> The demonstration of the evaluation results is not intuitive and straightforward.
>
> **Reply:**
> First, we apologize that the way we present the evaluation results is not intuitive. To facilitate understanding, we include a joint comparison in Section 5.1, where Figure A6 jointly compares the efficiency (throughputs) and effectiveness (accuracy) of different methods. Hopefully, it will further support our empirical analysis in Section 4 and Section 5.
>
> Best,
>
> Authors

---

### Meta-Review · Area_Chair_P2UJ · 2022-09-10

**Recommendation:** Accept
**Confidence:** 5

**Metareview:**

All the reviewers appreciate the extensive survey on training GNNs for large-scale graphs, the useful experimental observations, and the new ensemble-based training strategy (EnGCN) of this work. The main concerns were insufficient hyperparameter optimization, marginal novelty (compared to SIGN), lack of efficiency comparison, missing analysis on parameter sensitivity, and limited graph datasets in the experiments. In the rebuttal and discussion, the authors addressed most of the concerns raised, and the manuscript has been updated accordingly. AC thus recommends accepting this paper.

---

### Decision · Program_Chairs · 2022-09-16

Accept